# The Ectomycorrhizal Fungi and Soil Bacterial Communities of the Five Typical Tree Species in the Junzifeng National Nature Reserve, Southeast China

**DOI:** 10.3390/plants12223853

**Published:** 2023-11-14

**Authors:** Wenbo Pang, Panpan Zhang, Yuhu Zhang, Xiao Zhang, Yanbin Huang, Taoxiang Zhang, Bao Liu

**Affiliations:** 1College of Forestry, Fujian Agriculture and Forestry University, Fuzhou 350002, China; 18731112578@163.com (W.P.); zpanpan0819@163.com (P.Z.); zhangyuhu4790@163.com (Y.Z.); 2Key Laboratory of Soil Ecosystem Health and Regulation of Fujian Provincial University, College of Resources and Environment, Fujian Agriculture and Forestry University, Fuzhou 350002, China; zx2193447224@foxmail.com; 3Administration Bureau of Fujian Junzifeng National Nature Reserve, Mingxi 365200, China; jzfbhk@163.com

**Keywords:** forest ecosystems, tree species, microorganisms, ECM fungal community, soil bacterial community

## Abstract

To explore the contribution of microorganisms to forest ecosystem function, we studied the ectomycorrhizal (ECM) fungal and soil bacterial community of the five typical tree species (*Pinus massoniana*, PM; *Castanopsis carlesii*, CC; *Castanopsis eyrei*, CE; *Castanopsis fargesii*, CF; and *Keteleeria cyclolepis*, KC) at the Junzifeng National Nature Reserve. The results indicated that the ECM fungal and soil bacterial diversity of CC and CF was similar, and the diversity rates of CC and CF were higher than those of PM, CE, and KC. *Cenococcum geophilum* and unclassified_Cortinariaceae II were the most prevalent occurring ECM fungi species in the five typical tree species, followed by unclassified_Cortinariaceae I and *Lactarius atrofuscus*. In bacteria, the dominant bacterial genera were *Acidothermus*, *Bradyrhizobium*, *Acidibacter*, *Candidatus_Solibacter*, *Candidatus_Koribacter*, *Roseiarcus*, and *Bryobacter*. EMF fungi and soil bacteria were correlated with edaphic factors, especially the soil pH, TP, and TK, caused by stand development. The results show that the community characteristics of ECM fungi and bacteria in the typical tree species of the Junzifeng National Nature Reserve reflect the critical role of soil microorganisms in stabilizing forest ecosystems.

## 1. Introduction

Soil microbes are important drivers of biogeochemical cycling, acting as a key link between plants and ecosystems [1]. Ectomycorrhizal (ECM) fungi establish symbiotic associations, with approximately 60% of trees on Earth [2]. ECM fungi obtain carbon sources from host plants in exchange for water and soil nutrients that promote plant growth. These root-colonizing fungi increase plant growth and tolerance to stressful environments, such as high temperatures, heavy metals, and pathogen stress [2,3]. Soil bacteria represent the largest group of soil microorganisms and play crucial roles in regulating mycorrhizal symbiosis, driving nutrition cycle processes, maintaining biodiversity, suppressing pathogens, and protecting plant health [4]. The dynamic changes in soil microorganisms reflect the soil biochemical processes, plant health, and forest ecological functions [5]. Due to the widespread destruction of natural forests, the loss of microbial diversity has become a global concern in forestry [6]. Therefore, understanding the soil microbial community in stable natural forest ecosystems is key to maintaining the long-term stability of forest ecosystems.

Tree species affect ECM fungal communities through litter, root exudates, and root symbiotic microorganisms. Litter properties, including the quality of litter and specific chemical components released during litter decomposition, shape the ECM fungal diversity and community composition [7]. Several studies reported that ECM fungi with short exploration strategies were the major contributor to relatively recalcitrant leaf litter inputs [8]. Root exudates, including various compounds like organic acids, amino acids, sugars, and phenolics, have the potential to impact the ECM fungal community and diversity [9]. Glucosinolate and isothiocyanate secreted by Brassicaceae species inhibited mycorrhizal colonization, resulting in a different fungal community structure compared with the Asteraceae, Fabaceae, and Poaceae [9]. The phylogenetic traits and root architecture of trees also play significant roles in changing the ECM fungal community [10,11]. Host phylogeny explained 20% and 75% of the variation in the composition of the ECM fungal community and species richness in Salicaceae, respectively [10]. Differences in root morphology, such as root length and fine root density, as well as the below-ground allocation of the tree species, affect the ECM fungal community [10]. Trees with higher fine root density exhibit an increase in ECM species richness, which has greater nutrient acquisition potential to promote plant growth [10]. Moreover, tree species also affect the soil bacterial community. For example, *Picea abies* Karst (Pinaceae) and *Pinus sylvestris* L. (Pinaceae) acidify the soil through slow decomposition of litter, enriching the acidophilic bacterial taxa in the soil [12,13]. Fabaceous plants enrich *Rhizobium* in the rhizosphere by secreting isoflavonoids [9]. *Abies nordmanniana* Spach. (Pinaceae) and *P. abies* inhibit the abundance of nitrifying bacteria such as nitrobacterium and ammonia-oxidizing archaea by producing specific compounds [14].

Ectomycorrhizal fungi can provide nutrients for bacteria in soil by releasing secretions, and bacteria have strong chemotactic characteristics toward low-molecular-weight carbohydrates secreted by fungi [15]. Therefore, the symbiotic system of ECM fungi also indirectly promotes the selection of mycorrhizal growth-promoting bacterial populations [15]. At the same time, bacteria can colonize the hyphae of mycorrhizal fungi, promote mycelial growth, increase the infection rate of mycorrhizal plants, and promote plant growth and stress resistance [16]. Electron microscopy has shown that the ectomycorrhizospheres of *P. sylvestris*–*Suillus bovinus* (L. ex Fr.) 0. Kuntze (Suillaceae) and *P. sylvestris*–*Paxillus involutus* (Batsch ex Fr.) (Suillaceae) host distinct populations of bacteria [17]. Mycelial networks and carbon compounds secreted by fungal mycelia promoted bacterial migration and bacterial fitness in soil [18]. A previous study has revealed that fluorescent pseudomonads prefer to colonize the hyphae of the mycorrhizal fungus *Laccaria bicolor* S238N (Hydnangiaceae) [18]. On the other hand, ECM fungi can influence the community structure of mycorrhizal bacteria via the exudations from mycelia, such as amino acids, peptides, and sugars [19]. A previous study showed that *Pseudomonas* species in the mycosphere utilized specific fungal exudates involving L-leucine, D-mannitol, L-arabinose, m-arabitol, and D-trehalose to increase plant growth and resistance against plant pathogens [19]. In return, bacteria stimulate the germination of fungal propagules, promote mycelial growth, facilitate mycorrhizal colonization, and antagonize pathogenic fungi [15]. *Pseudomonas fluorescens* BBc6R8 (Pseudomonadaceae) has been shown to stimulate the growth of the ECM fungus and induce morphological changes in the hyphal, which may be beneficial for the formation of mycorrhizal symbiont [20]. The bacterium *Paraburkholderia terrae* BS001 can detoxify antifungal compounds and occupy possible niches that may be occupied by pathogens, thus helping to protect fungal species [21]. However, a number of mycorrhizal fungi might compete with bacteria for niche by producing defensins [22,23]. Shirakawa et al. found that *S. bovinus* exhibited antibacterial activity against Gram-positive bacteria in the mycorrhizosphere [23].

The Junzifeng National Nature Reserve is located in Fujian Province, China. It is the only primeval subtropical evergreen broad-leaved forest with a large area (1730 km^2^) and low-latitude distribution [24]. The Junzifeng National Nature Reserve has an extremely rich biodiversity, with a total of 219 families and 779 genera of vascular plants [25]. Due to the large geographic area and its rich biodiversity, the reserve is conducive to combatting climate change, maintaining global carbon balance, and protecting biodiversity and stable maintenance of the ecosystem [25]. Previous studies have reported that this reserve could sequester over 1.1 lakh tons of carbon per year [26]. This carbon sequestration function of forests is beneficial for stabilizing and even reducing the concentration of greenhouse gases [27]. Liu et al. found a significant positive correlation between the dominant tree species of the *Michelia odora* (Magnoliaceae) community in the Junzifeng National Nature Reserve, indicating that the community was at a relatively stable succession stage [28], which provides significant insight for understanding the stability of forest communities. However, few researchers have recognized the role that microbes play in the processes. Therefore, this research selected five typical tree species from the reserve, namely, *Pinus massoniana* Lamb (Pinaceae) (PM); *Castanopsis carlesii* (Hemsl.) Hayata (Fagaceae) (CC); *Castanopsis eyrei* (Champ. ex Benth.) Hutch. (Fagaceae) (CE); *Castanopsis fargesii* Franch. (Fagaceae) (CF); and *Keteleeria cyclolepis* Flous (Pinaceae) (KC). The present study aimed to (1) characterize the ECM fungal/bacterial community diversity and composition of the five typical tree species; (2) explore the potential correlation between ECM fungi and soil bacteria; and (3) explore the role of fungi and bacteria in the biodiversity and forest ecosystem stability of the Junzifeng National Nature Reserve. Ultimately, our findings emphasize the role of soil microorganisms in stable forest ecosystems, improve understanding of the ecosystem function of soil fungi and bacteria in subtropical forests, and provide insight into sustainable forest management.

## 2. Results

### 2.1. ECM Fungal Diversity and Community Composition in the Five Tree Species

Figure 1 shows 18 ECM morphotypes in the 20 root samples collected from CC, PM, CE, CF, and KC. After sequence analysis, 18 ECM fungal taxa were identified (Appendix A). Of these 18 taxa, 13 belong to various genera, including Amanita, Cenococcum, Cortinarius, Hortiboletus, Helotiales, Inocybe, Lactarius, Russula, Sebacina, and Tomentella. Additionally, five taxa from the families Cortinariaceae, Trichocomaceae, and Thelephoraceae were identified (Appendix A). The species richness values of ECM fungi in CC, PM, CE, CF, and KC were nine, six, four, four, and three, respectively (Appendix A).

Shannon–Wiener indices in CC and CF were higher than in PM, CE, and KC, and they were significantly higher than that in CE (Figure 2a). Simpson and Pielou evenness indices were not significantly different among the five tree species (Figure 2b,c), but the indices in CC and CF were higher than those in PM, CE, and KC. The two tree species with the greatest similarity to ECM fungal composition were CC–CF (S = 0.46, J = 0.19) (Appendix A).

Based on the importance value (IV) shown in Figure 3 and Appendix A, *Cenococcum geophilum* (24.7%), *Cortinariaceae* sp. II (15.75%), *Inocybe posterula* (19.08%), *Lactarius vividus* (19.9%), and *Thelephoraceae* sp. (12.3%) were common species in PM. In CC, *Cortinariaceae* sp. II (34.3%), *C. geophilum* (24.08%), and *Hortiboletus rubellus* (10.22%) were the common species. In CE, *Cortinarius carneoroseus* (11.82), *Sebacina* sp. (11.82), and *Lactarius atrofuscus* (10.42%) were the common species, and *Cortinariaceae* sp. II (65.94%) was the dominant species. In CF, *Russula xerampelina* (42.61%), *C. geophilum* (33.79%), and *Cortinariaceae* sp. I (18.27%) were the common species. In KC, *Russula compacta* and *Tomentella* sp. were the common species, and *C. geophilum* (53.48%) was the dominant species. The most prevalent mycorrhizal species among the five tree species were *C. geophilum* and unclassified_Cortinariaceae II, both of which were recorded across four out of the five tree species. The highest number of specific species were found in CC, including Amanita pseudosychnopyramis, *Trichocomaceae* sp., *Russula aff. kansaiensis*, *Russula* sp., *Helotiales* sp., and *Hortiboletus rubellus*.

### 2.2. Bacterial Diversity and Community Composition in the Five Tree Species

A total of 1,311,618 effective sequences and 4105 OTUs were obtained at a 97.0% similarity level. The slopes of dilution curves of all samples were close to saturation (Appendix A). The number of shared OTUs among PM, CC, CE, CF, and KC was 1195. The unique OTUs followed the order of CF (344) > CC (253) > CE (226) > PM (211) > KC (193) (Appendix A).

The Chao1 in CF was higher than in PM, CC, CE, and KC, and significantly higher than that in CE (Table 1). The ACE indices showed similar trends, with CF having the highest diversity. However, there were no statistical differences in Simpson and Shannon indices among the five tree species (Table 1). Furthermore, the NMDS analysis revealed that the bacterial community of CF and CC clustered together, while CE, PM, and KC clustered together (Figure 4), indicating that the bacterial community compositions of CF and CC were similar, and those of CE, PM, and KC were also highly similar.

Figure 5 shows the relative abundances of bacterial phyla and genera. We detected 48 bacterial phyla, among which Proteobacteria (16.16–23.17%), Acidobacteria (18.26–23.75%), Actinobacteria (8.60–9.75%), WPS-2 (0.87–3.75%), and Chloroflexi (1–2.88%) were dominant phyla (with an average relative abundance > 1%) (Figure 5 and Appendix A). The relative abundance rates of Proteobacteria and Acidobacteria were highest in CF (23.17% and 23.75, respectively) and lower in PM (16.16% and 21.47%, respectively) and KC (21.72% and 18.26%, respectively). The relative abundance of WPS-2 was significantly higher in CE (5.29%) than in CC (1.38%), KC (1.47%), and CF (0.87%). The relative abundance of Chloroflexi in CE (2.88%) and CF (2.37%) was significantly higher than in PM (1.21%) and KC (1.00%).

At the genus level, the dominant soil bacteria (with an average relative abundance > 1%) included *Acidothermus* (5.85%), *Bradyrhizobium* (4.41%), *Acidibacter* (4.06%), *Candidatus_Solibacter* (3.80%), *Candidatus_Koribacter* (3.43%), *Roseiarcus* (2.22%), and *Bryobacter* (2.18%) (Figure 5 and Appendix A). There were no significant differences in the relative abundance of *Acidibacter*, *Candidatus_Koribacter*, *Roseiarcus*, and *Bryobacter* among the tree species (Appendix A). *Acidothermus* was more abundant in PM (6.94%) and KC (7.43%) than in CC (5.31%), CE (6.03%), and CF (3.53%). The relative abundance of *Bradyrhizobium* in PM (2.82%) was significantly lower than in CC (5.58%), and the relative abundance of *Candidatus_Solibacter* in PM (3.24%) was also significantly lower than in the other tree species.

LEfSe analysis identified the specific bacteria whose relative abundance significantly varied among the five tree species (Figure 6). We found that CF had the most different genera, including *Candidatus*_*Solibacter*, unidentified_*Alphaproteobacteria*, *Sphingomonas*, *Niastella*, *Reyranella*, *Roseomonas*, *Jatrophihabitans*, and *Rhodopseudomonas*. *Granulicella, Candidatus_Xiphinematobacter*, and *Eubacterium_hallii_group* were enriched in CC. *Pseudolabrys* and *Pajaroellobacter* were enriched in PM. *Acidicaldus* was enriched in CE, and *Subdoligranulum* was enriched in KC.

### 2.3. Effect of Soil Properties on ECM Fungal and Bacterial Communities

The RDA explained 50.21% and 52.64% of the total variation in the ECM fungal community (Figure 7a) and bacterial community (Figure 7b), respectively. For the ECM fungal community, pH, TP, TK, and DON were significantly related to the two axes, with pH being the primary factor (R^2^ = 0.73), followed by TP (R^2^ = 0.66) (Appendix A). For the bacterial community, soil properties, except DON, were significantly related to the two axes, with TK being the primary factor (R^2^ = 0.94), followed by pH (R^2^ = 0.74) (Appendix A).

The relationship between soil properties and microorganisms at the genus level was analyzed using the Pearson correlation analysis (Figure 8a,b). For ECM fungi, *Cortinarius, Cortinariaceae* sp., and *Sebacina* were significantly positively correlated with dissolved organic nitrogen (DON), and *Cortinarius* was negatively correlated with pH (Figure 8a). *Lactarius* was significantly positively correlated with soil organic matter (SOM). *Russula* was significantly positively correlated with pH and total potassium (TK) while negatively correlated with dissolved organic carbon (DOC). *Tomentella* was significantly positively correlated with TP. The Shannon index was significantly positively correlated with pH and TK while negatively correlated with total phosphorus (TP) and DON. In terms of the bacterial community, *Acidothermus* was significantly positively correlated with SOM, TP, and DOC while negatively correlated with pH and TK (Figure 8b). *Bradyrhizobium* was positively correlated with pH, and *Bryobacter* was positively correlated with SOM and DOC. *Candidatus*_*Koribacter* was significantly positively correlated with pH while negatively correlated with DOC. *Candidatus*_*Solibacter* was significantly positively correlated with pH and TK while negatively correlated with SOM. *Roseiarcus* was significantly positively correlated with TP while negatively correlated with pH. The Chao1 and ACE indices were significantly positively correlated with TK and pH, while the ACE index was negatively correlated with TP.

## 3. Discussion

Forest trees are associated with hundreds of microorganisms, among which ECM fungi and soil bacteria contribute to the maintenance and functioning of forest ecosystems. Currently, the loss of soil microbial diversity has disrupted the balance and stability of forest ecosystems [29]. For example, the conversion of natural forests into forest plantations has homogenized microbial communities and led to a decrease in microbial diversity [30]. The diversity and abundance of potassium-solubilizing bacteria in plantation soil are lower than in forest soil, implying that the available K in plantation decreases, which leads to soil acidification and reduced forest productivity and pathogen resistance [31]. The Junzifeng National Nature Reserve is the only primeval subtropical evergreen broad-leaved forest with a large area, which plays a key role in providing biodiversity and functional diversity, as well as maintaining forest ecosystem stability [24,25]. Therefore, to understand the role of soil microorganisms in maintaining stable forest ecosystems, it is critical to explore soil microbial species and their ecological function among different tree species in forest ecosystems. In this study, we compared the ECM fungal and soil bacterial communities of five typical tree species (PM, CC, CE, CF, and KC) in the Junzifeng National Nature Reserve to explore the functional roles of ECM fungi and soil bacteria in forest ecosystem stability.

In our study, the microbial community in CC and CF was similar (Figure 4 and Appendix A), and the microbial diversity of CC and CF was higher than that of PM, CE, and KC (Figure 2, Table 1). CC and CF are typical broad-leaved tree species in subtropical natural forests. The analysis of phylogenetic relationships among plants indicates that host species within the same plant genus or family tend to support similar ECM communities [10]. Compared with PM, CE, and KC, CC and CF mainly affect microbial diversity in terms of three aspects. First, Fagaceae generally have an extensive production of fine roots than a Pinaceae plant [32], which may provide more rhizosphere areas to accommodate microbes, thus supporting diverse community characteristics. Therefore, CC and CF may have more fine roots than PM and KC, which leads to higher microbial diversity. Second, compared with broad-leaved trees, conifers return fewer cations to the soil and produce more organic acids during the slow decomposition of litter, leading to an increase in soil acidity [33]. Hüblová and Frouz found higher soil pH in broad-leaved forests than in coniferous forests in both natural and plantation forest soils [34]. Dawud et al. [33] proved that soil pH decreased with an increase in the dominance of conifer species in major European forest types. The long-term differences in tree species have resulted in variations in soil pH, subsequently exerting a significant influence on ECM communities among forests [35]. According to Rosinger et al. [36], the European beech, pine, and spruce forests exhibited the highest levels of ECM diversity in soils with pH close to neutral. Hence, the difference in pH among different tree species caused by litter may result in higher microbial diversity of CC and CF than that of PM and KC. Third, research has shown that leaf nutrient content and litter decomposition rate in broad-leaved forests are generally higher than those in coniferous forests, directly promoting the accumulation of nutrients in the soil of broad-leaved forests and increasing the soil microbial diversity [37]. Our research yielded similar results, where a significant positive correlation between TK and microbial diversity was observed. In our study, soil TP was negatively correlated with microbial diversity. Studies from China have shown that soil TP concentrations below 0.4 g·kg^−1^ are indicative of a possible shortage of phosphorus supply in the soil [38]. In this study, soil total P concentrations of the CC and CF were 0.11 and 0.16, respectively. Other studies have confirmed that the growth of some ECM fungi is promoted with decreased P levels under pure culture conditions [39]. Therefore, the difference in leaf nutrient content between broad-leaved and coniferous forests may be an important factor that leads to higher ECM diversity in CC and CF than in PM and KC.

*Cenoccocum geophilum* and *Cortinariaceae* sp. II were the most frequently occurring ECM species in the five typical tree species (Figure 3), and the importance values of the two ECM fungi were generally higher than other ECM fungi (Figure 3). Consistent with our study, some studies revealed that *C. geophilum* and Cortinariaceae were frequently observed in natural forest ecosystems [40]. *Cenoccocum geophilum*, a common ECM fungus in natural forests, is capable of establishing ECM associations with a diverse range of host trees, including Pinaceae, Fagaceae, and Betulaceae [41]. The inoculation of *C. geophilum* in Masson pine seedlings resulted in enhanced plant growth and stress resistance through improved biomass production, photosynthesis, and nutrient status [41]. Our previous research confirmed that *C. geophilum* exhibited resistance to drought, high temperature, and heavy metal stress by increasing nutrient absorption and antioxidant enzyme activity and upregulating genes encoding antioxidant enzyme activity, ubiquinone, and other terpenoid quinones and metabolism of compounds [42,43,44]. Additionally, our research demonstrated that the inoculation of *C. geophilum* effectively alleviated heavy metal-induced toxicity in *Pinus massoniana* seedlings [45]. Wu et al. [46] conducted a study on the distribution of *C. geophilum* in zonal vegetation in Inner Mongolia and found that the environment with stable stand, high soil moisture, and rich organic matter content was conducive to the colonization and growth of *C. geophilum*. Cortinariaceae is a key player in the mobilization of nutrients, which is linked to nutrient cycling in forest ecosystems [47]. In eucalypt forests, Cortinariaceae has great richness in healthy forests [48]. However, plantations generally have a significant loss of microbial diversity compared with natural forests, which may lead to fragile forest ecosystems [49]. According to a study investigating the ECM fungal community composition of *Pinus massoniana* plantation in summer, *Russula* (43.63%), *Amanita* (19.40%), *Lactarius* (19.04%), *Tomentella* (7.31%), *Clavulina* (5.25%), and *Amphinema* (4.76%) were the dominant genera, while *C. geophilum* was detected as a rare genus (0.3%), and the species of Cortinariaceae was not detected [50]. Yang et al. [51] examined the ECM community in another *P. massoniana* plantation in southwestern China and found that *Russula* and *Tomentella* were the dominant genera, while *C. geophilum* and *Cortinariaceae* sp. II were not detected. Therefore, we speculate that *C. geophilum* and *Cortinariaceae* sp. II might be more prevalent in natural forests than in plantations, reflecting their critical role in maintaining the stability of forest ecosystems. Hence, *C. geophilum* and Cortinariaceae could be potential candidate microorganisms to be used in biofertilizers to improve the ecological stability of plantations and restore declining forests.

Proteobacteria, Acidobacteria, and Actinobacteria were the dominant soil bacterial phyla of the five tree species in our study (Figure 5 and Appendix A), which are supported by some previous studies [52,53]. Urbanova et al. [52] found that Proteobacteria, Actinobacteria, and Acidobacteria accounted for 50.7%, 16.7%, and 8.4% of the forest soil microbial communities, respectively. Proteobacteria are capable of decomposing soil organic matter, fixing nitrogen and carbon, and mobilizing phosphorus [54]. Previous research highlighted the ubiquity and species diversity of the Proteobacteria in forest ecosystems [52]. Actinobacteria is a flora that utilizes a broad range of carbon sources and secretes some secondary metabolites to promote plant growth and defense against pathogen attack [55,56]. In our study, Acidobacteria was the dominant bacterial phylum, which is consistent with the widespread distribution of Acidobacteria in subtropical regions [52]. In addition, the differences in tree characteristics among the five tree species resulted in the enrichment of some specific soil bacteria. Specifically, the relative abundance of *Candidatus solibacter* in CC and CF was higher than that in PM and KC (Appendix A and Figure 5b). *C. solibacter* is an aerobic ammonia-oxidizing bacterium (AOB), which participates in the reduction of NO_3_^−^ and NO_2_^−^ and the decomposition of organic carbon by producing enzymes [57]. Li et al. [58] found that, in comparison to coniferous forests, broad-leaved forest soils exhibit a dominance of AOB abundance with relatively high gross and net nitrification rates, and the abundance in natural forests is higher than that in artificial forests. Therefore, the input of organic matter and nitrification potential may have led to the enrichment of *C. solibacter* in CC and CF. In addition, *C. solibacter* is a mycorrhiza-promoting bacterium, which is capable of interacting with EMF species to promote mycorrhizal colonization and plant growth [59]. Gui et al. [59] observed that the abundance of *C. solibacter* is influenced by the mycorrhizal fungus *Funneliformis mosseae*. In the present study, the importance values of *C. geophilum* in CC and CF were generally higher than other ECM fungi (Figure 3). It was reported that *C. geophilum* had relatively high phenoloxidase activity, which promoted the content of soil inorganic carbon [60]. Zhang et al. [61] suggested that the soil carbon content was positively correlated with the relative abundance of *C. solibacter*. Therefore, we speculate that certain mycorrhiza-promoting bacteria, such as *C. solibacter*, may be enriched in the rhizosphere of *C. geophilum*. We also found a significant enrichment of *Sphingomonas* in CF (Figure 6), which may be associated with exudates from ECM fungi, such as arabinose, inositol, mannitol, and trehalose [62]. Studies have indicated that compounds produced by *Russula*, such as sugar alcohols, inositol, mannitol, and trehalose, can enrich some species of *Sphingomonadaceae* in the rhizospheres [62]. Therefore, there exists a mutually beneficial relationship among tree species, ectomycorrhizal fungi, and soil bacteria, which contributes to forest nutrient cycling, plant disease resistance, and forest stability.

## 4. Materials and Methods

### 4.1. Study Area

This study was carried out in the Junzifeng National Nature Reserve (116°47′–117°31′ E and 26°19′–26°39′ N) in southeastern China, located in a subtropical region with a total area of 180.7 km^2^. The reserve has a mid-subtropical marine monsoon climate, with an annual rainfall of 1737 mm and an annual temperature of 18 °C. Five tree species were included in this study: *Pinus massoniana* (PM), *Castanopsis carlesii* (CC), *Castanopsis eyrei* (CE), *Castanopsis fargesii* (CF), and *Keteleeria cyclolepis* (KC). Characteristics of the sampling sites are listed in Appendix A.

### 4.2. Sample Collection and Processing

In each of the five forest sites, four sample plots of 20 m × 20 m were set. Within each plot, three standard trees with an average breast-height diameter were selected and the distance between the trees was >10 m. After removing herbs, litter, and humus layer, fine roots (≤2 mm in diameter) and rhizosphere soil were collected at a soil depth of 0–30 cm in four directions of each standard tree and were merged to form a root sample and rhizosphere soil sample. Then, the samples of three standard trees were mixed in each plot to form a composite sample. In total, 40 samples of roots and rhizosphere soils were stored, placed in an ice box, and then immediately transported to the laboratory. In the laboratory, the root samples were gently cleaned and then stored at 4 °C until the identification of the ECM fungi. The soil was divided into two halves: one half was used for DNA extraction, which was stored at −80 °C, while the other half was used for soil physicochemical analyses, which was air-dried and sieved.

### 4.3. Soil Property Analysis

Soil properties were analyzed following the Agricultural Chemistry Committee of China [38]. Soil pH was measured using a soil-to-water ratio of 1:5 with an electronic pH meter (Ohaus ST2200-F, Parsippany, NJ, USA). Soil organic matter (SOM) content was measured using the potassium dichromate volumetric method. Total phosphorus (TP) and total potassium (TK) were digested with HF-HClO_4_ and determined using the molybdenum blue method and flame photometry, respectively. The dissolved organic carbon (DOC) concentration and dissolved organic nitrogen (DON) were extracted with a soil–water ratio of 1:5, and the concentration was measured with a total organic carbon analyzer (DOC-L CPH, Shimadzu; Tokyo, Japan). Appendix A shows the detailed soil properties.

### 4.4. Identification and Data Analysis of ECM Fungi

Roots were carefully washed under running tap water in a set of sieves (2 and 0.149 mm) and cut into 2–3 cm pieces. Briefly, 3000 root tips of each sample were randomly observed under a stereomicroscope (M205 FA, Leica, Bensheim, Germany). After preliminary classification, the root tip number of each type of mycorrhizal fungi was recorded, and then the typical ECM fungi were photographed using the stereomicroscope [63]. Three replicate ECM root tips of each morphotype were cleaned, placed individually in 2.0 mL tubes, and stored at −20 °C.

DNA of each morphotypical ECM fungi was extracted using the cetyl trimethyl ammonium bromide (CTAB) method [64]. The samples were amplified using ITS-1F and ITS-4 primers [65], and the qualified PCR products were sent to Beijing Tsingke Biotech Co., Ltd., for sequencing. The obtained sequence data of ECM fungi were edited and manually inspected and edited using SnapGene Viewer (version 7.0.8) and MEGA (version 5) software, and then BLAST alignment was performed in the NCBI database. The sequences were deposited in NCBI-SRA (Accession Numbers: OR258990–OR259007). A match at a similarity of ≥97% represents a species, ≥90% represents a genus, and ≥80% represents a family [63]. To study the alpha diversity of the ECM fungal communities, measures such as Shannon, Simpson, and Pielou indices were used [66]. Sorensen (S) and Jaccard (J) similarity indices were used to evaluate the similarity of ECM communities among the tree species [66]. The relative frequency (RF) and relative abundance (RA) of an ECM fungus were defined as the number of soil samples in which that species occurred and the percentage of ECM tips colonized by that fungus out of the total number of ECM tips observed, respectively [63]. Furthermore, the importance value (IV) of ECM fungi was determined by calculating the average of the RF and RA. A dominant fungus is defined as a species with an IV ≥ 50, a common fungus is defined as a species with 10% < IV < 50%, and a rare fungus is defined as a species with an IV ≤ 10% [67].

### 4.5. DNA Extraction and Data Analysis of Soil Bacteria

Soil DNA extraction, amplification, and sequencing were performed at Novogene Bioinformatics Technology Co., Ltd. (Beijing, China). DNA of soil samples was extracted using the CTAB method. The primers 338F and 806R were used to amplify the V3-V4 variable region, and the PCR products were purified with a Qiagen Gel Extraction Kit (Qiagen, Hilden, Germany). Libraries were constructed with a Truseq DNA PCR-Free Sample Preparation Kit (Illumina, San Diego, CA, USA). The constructed libraries were analyzed with a Qubit^®^ 2.0 fluorometer (Thermo Scientific, Waltham, MA, USA) and the Agilent Bioanalyzer 2100 system, and sequenced with the NovaSeQ 6000 system. The sequences were clustered at 97% similarity using Uparse software (Uparse v7.0.1001). The rarefaction curves and Venn diagrams were analyzed based on the OTU abundance data using the R software package (Version 2.15.s3). Alpha diversity indices, including ACE, Chao1, Shannon, and Simpson indices, were used to reflect the bacterial diversity among the five tree species [50]. Based on the Bray–Curtis similarity matrix, NMDS was conducted to show the differences and similarities of the soil bacterial distributions.

### 4.6. Statistical Analyses

Data analysis was performed using Microsoft Excel 2020 and SPSS 26.0, and figures were created using Origin 2022. The data were described as means of four replicates plus or minus one standard deviation. The significant differences in the ECM fungal and soil bacterial communities among the five tree species were analyzed using ANOVA and Tukey’s HSD tests (*p* < 0.05). The redundancy analysis (RDA) was performed via Canoco 5, to evaluate the linkages between main ECM fungal and bacterial communities related to soil properties. The Pearson correlation coefficient between soil properties and ECM fungal communities, as well as the correlation between soil properties and bacterial communities, were calculated in SPSS 26.0, with *p* < 0.05, *p* < 0.01, and *p* < 0.001 chosen as the levels of significance.

## 5. Conclusions

According to the findings of our study, differences in characteristics between broad-leaved and coniferous tree species resulted in similar ECM and soil bacterial diversity between CC and CF and significantly higher diversity in CC and CF than in PM and KC. Furthermore, the enrichment of specific bacteria in CC and CF may be associated with secondary metabolites produced by ECM fungi. The beneficial interactions among tree species, ECM fungi, and soil bacteria contributed to the mobilization of soil nutrients, plant growth, and stress resistance in the stable forest ecosystem. In the microbial communities of the reserve, ECM fungi (*C. geophilum* and unclassified_Cortinariaceae II) and soil bacteria (*C. solibacter*) have the potential to serve as beneficial plant inoculants to enhance forest stability in complex environments, including plantation and declining forests.

## Figures and Tables

**Figure 1 plants-12-03853-f001:**
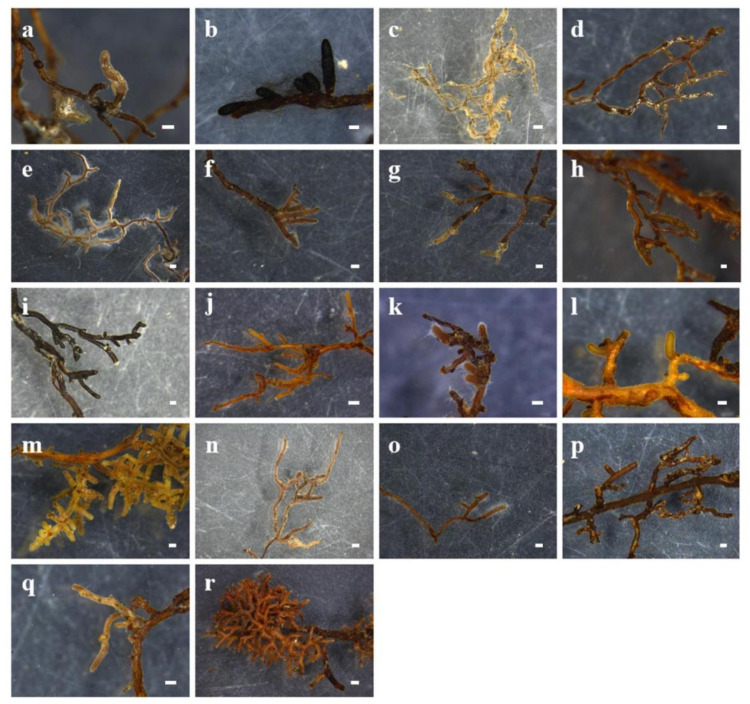
Morphological features of different ECM fungi: (**a**) *Amanita pseudosychnopyramis* in CC; (**b**) *Cenococcum geophilum* in PM, CC, CF, and KC; (**c**) *Cortinarius carneoroseus* in CE; (**d**) *Cortinariaceae* sp. I in CC and CF; (**e**) *Cortinariaceae* sp. II in PM, CC, CE, and CF; (**f**) *Hortiboletus rubellus* in CC; (**g**) *Helotiales* sp. in CC; (**h**) *Inocybe posterula* in PM; (**i**) *Lactarius atrofuscus* in PM and CE; (**j**) *Lactarius vividus* in PM; (**k**) *Russula aff. Kansaiensis* in CC; (**l**) *Russula compacta* in KC; (**m**) *Russula xerampelina* in CF; (**n**) *Russula* sp. in CC; (**o**) *Sebacina* sp. in CE; (**p**) *Tomentella* sp. in KC; (**q**) *Trichocomaceae* sp. in CC; (**r**) *Thelephoraceae* sp. in PM; bars = 300 μm.

**Figure 2 plants-12-03853-f002:**
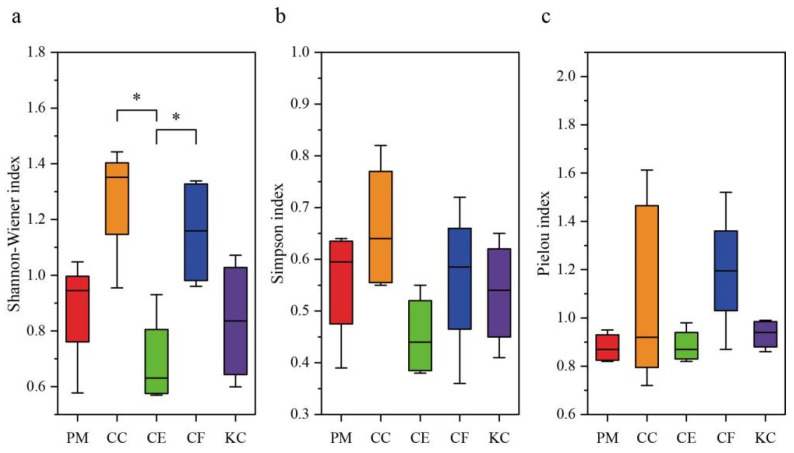
ECM diversity of the five tree species in Shannon-Wiener index (**a**), Simpson index (**b**) and Pielou’s index (**c**). *Pinus massoniana* (PM), *Castanopsis carlesii* (CC), *Castanopsis eyrei* (CE), *Castanopsis fargesii* (CF), and *Keteleeria cyclolepis* (KC); * values indicate significant differences (*p* < 0.05) among five tree species based on one-way ANOVA followed by Tukey test.

**Figure 3 plants-12-03853-f003:**
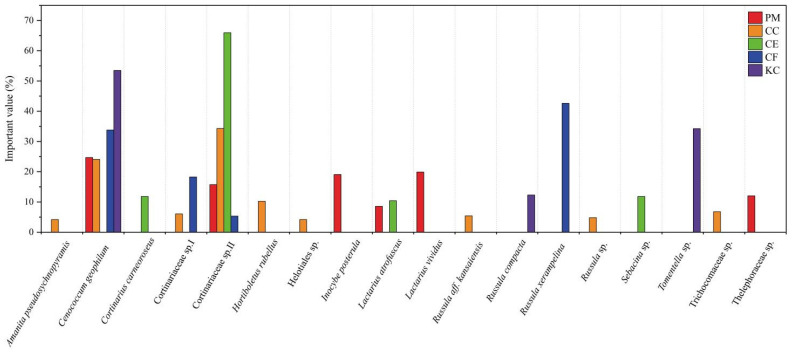
Comparison of importance values for ectomycorrhizal species on five tree species. *Pinus massoniana* (PM), *Castanopsis carlesii* (CC), *Castanopsis eyrei* (CE), *Castanopsis fargesii* (CF), and *Keteleeria cyclolepis* (KC).

**Figure 4 plants-12-03853-f004:**
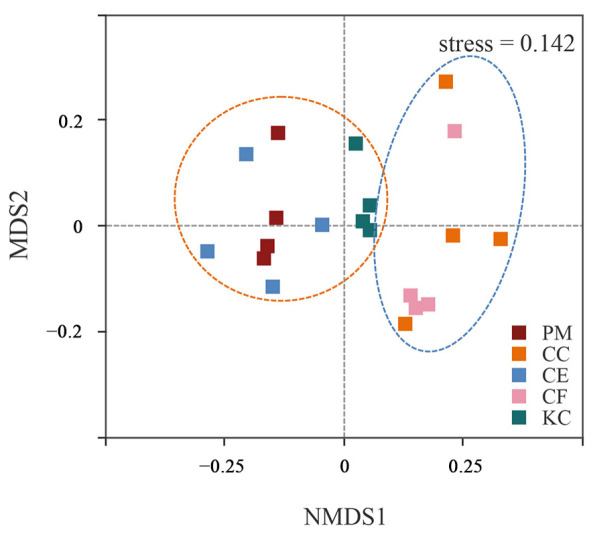
Non−metric multidimensional scaling (NMDS) plots of the bacterial communities based on Bray–Curtis dissimilarities. *Pinus massoniana* (PM), *Castanopsis carlesii* (CC), *Castanopsis eyrei* (CE), *Castanopsis fargesii* (CF), and *Keteleeria cyclolepis* (KC).

**Figure 5 plants-12-03853-f005:**
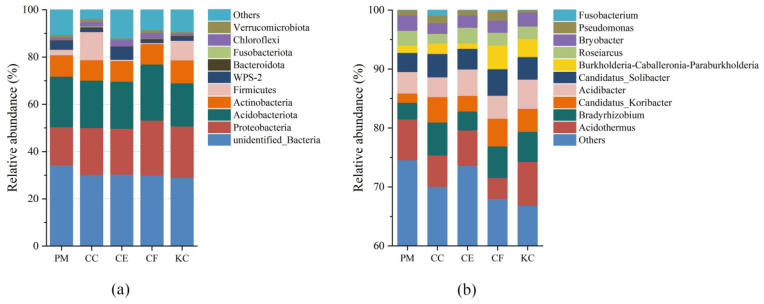
The top 10 most abundant bacteria at the phylum level (**a**) and genus level (**b**) in soil samples among the five tree species. *Pinus massoniana* (PM), *Castanopsis carlesii* (CC), *Castanopsis eyrei* (CE), *Castanopsis fargesii* (CF), and *Keteleeria cyclolepis* (KC).

**Figure 6 plants-12-03853-f006:**
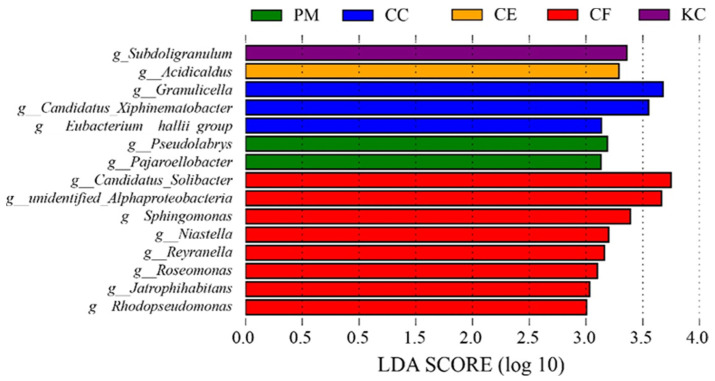
LEfSe analysis showing the bacterial genera that were significantly different among different tree species. *Pinus massoniana* (PM), *Castanopsis carlesii* (CC), *Castanopsis eyrei* (CE), *Castanopsis fargesii* (CF), and *Keteleeria cyclolepis* (KC).

**Figure 7 plants-12-03853-f007:**
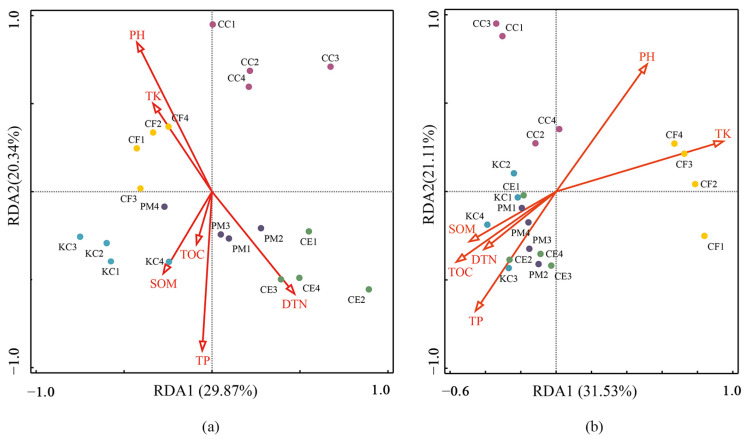
Redundancy analysis (RDA) for soil properties and ECM fungal genera (**a**), and bacterial genera (**b**) among the five tree species. *Pinus massoniana* (PM), *Castanopsis carlesii* (CC), *Castanopsis eyrei* (CE), *Castanopsis fargesii* (CF), and *Keteleeria cyclolepis* (KC).

**Figure 8 plants-12-03853-f008:**
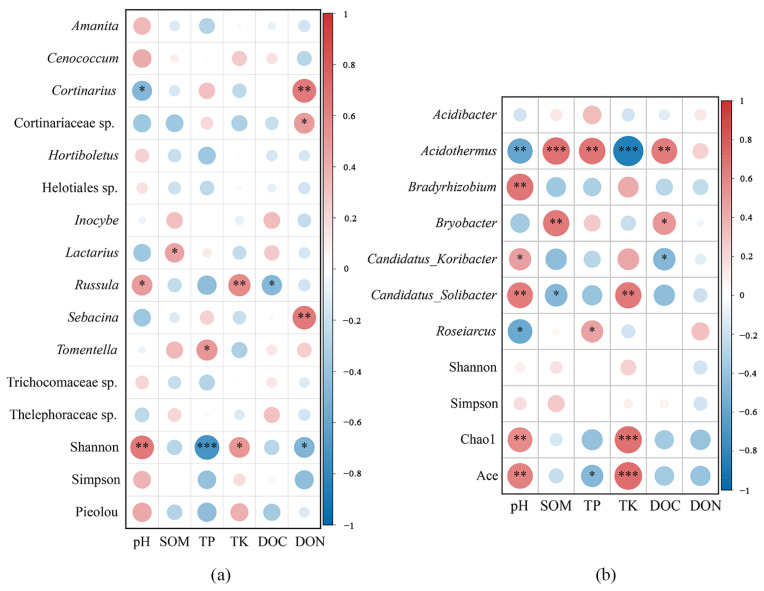
Pearson’s correlation coefficients between ECM fungi at the phylum level (**a**) and bacteria at the genus level (**b**) in communities and soil parameters among the five tree species; soil potential hydrogen (pH), soil organic matter (SOM), total phosphorus (TP), total potassium (TK), dissolved organic carbon (DOC), and dissolved total nitrogen (DON) were analyzed; * *p* < 0.05; ** *p* < 0.01 and *** *p* < 0.001.

**Table 1 plants-12-03853-t001:** Soil bacterial community diversity among the five tree species.

Tree Species	Chao1 Index	ACE Index	Simpson Index	Shannon Index
PM	1637.22 ± 155.21 ab	1647.12 ± 120.8 b	0.99 ± 0 a	7.96 ± 0.2 a
CC	1654.06 ± 127.23 ab	1695.17 ± 150.16 ab	0.99 ± 0 a	7.73 ± 0.29 a
CE	1552.47 ± 136.08 b	1587.25 ± 141.19 b	0.99 ± 0.01 a	7.86 ± 0.19 a
CF	1898.41 ± 118.04 a	1915.39 ± 68.05 a	0.99 ± 0 a	8.02 ± 0.12 a
KC	1646.99 ± 69.74 ab	1665.88 ± 50.9 b	0.99 ± 0 a	7.85 ± 0.23 a

Values are the average of four repeats and error bars indicate the standard deviations. *Pinus massoniana* (PM), *Castanopsis carlesii* (CC), *Castanopsis eyrei* (CE), *Castanopsis fargesii* (CF), and *Keteleeria cyclolepis* (KC); different letters indicate significant differences at level *p* < 0.05.

## Data Availability

The data presented in this study are available within the article and Appendix A.

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
