# Peer review of "The Ectomycorrhizal Fungi and Soil Bacterial Communities of the Five Typical Tree Species in the Junzifeng National Nature Reserve, Southeast China"

_plants, 2023, doi:10.3390/plants12223853_

Round 1

Reviewer 1 Report

Comments and Suggestions for Authors

Dziękuję za zaproszenie na spotkanie z manuskryptem zatytułowanym:

Ten Grzyby ektomikoryzowe i zbiorowiska bakterii glebowych pięciu typowych drzew Gatunki w Narodowym Rezerwacie Przyrody Junzifeng, południowo-wschodnie Chiny

Autorzy zdefiniował cel obserwacji i postawił problem badawczy.

Ten materiał i metody są wystarczająco opisane. Analizy statystyczne są wystarczająco dobrane.

Rezultaty są odpowiednio omówione.

Ten literatura jest odpowiednio dobierana.

Komentarze Sugestie dotyczące manuskryptu są następujące:

Zamówienie rozdziałów. Raczej materiał i metody powinny poprzedzać Wyniki i dyskusja.

Techniczny Korekty:

-Uzasadniać tekst w każdym rozdziale.

- L301 oraz L305 rozpocznij zdanie od całej nazwy łacińskiej

- L309 ".... stan odżywienia[43]". Jest zbędny przecinek

- L281 zbędna przestrzeń

- L286 "... zbliżony do neutralnego. [38]." Czy literatura ma do tego zastosowanie? zdanie? To po co ten okres?

- L294 "0,4 g kg-1 " wstawić symbol mnożenia

- L285 "Rosinger et al.", podaj cytaty w nawiasach dla literatura

- L281 "Vesterdal et al." podają cytaty z literatury w Nawiasy

- L326 "Yang et al." podają w nawiasach cytaty z literatury

Zamówienie cytowania/numeracji literatury, np. pozycja [70] L441 znajduje się w sekcji 4.5 i jest to sekcja przed Wyniki, Dyskusja

Jakość i rozdzielczość wykresów do poprawy.

Proponuję używać kolorów na wykresie 3, jest nieczytelny.

Dla tabele, proponuję skrócić podpis, aby nie powtarzać informacji z Materiał i metoda:

Różny litery wskazują na istotne różnice (P < 0,05) pomiędzy typami lasów 176 w oparciu o jednoczynnikową ANOVA, po której następuje test Tukeya. = Różne litery wskazują na istotne różnice na poziomie (P < 0,05).

Proszę ujednolicić zapis w nawiasach w cytatach dla rysunku lub ryc.

Po prostu daj na przykład (rys. 5, tabela S6), a nie tak jak jest (rysunek 5 i Tabela uzupełniająca S6).

Do stołu Rys.1, Tabela S6, Tabela S7, proponuję podać wartości Anova statystyka (F) i wartości prawdopodobieństwa dla P, w dodatkowych wierszach.

Z uwagi na fakt, że autorzy stwierdzają w Material and Methods, że przeprowadzili analizy w p<0,05, proponuję na rys. 8 wskazać tylko znaczenie dla p<0,05. Nie rozdzielaj już wyników prawdopodobieństwa dla progów 0,01 i 0,001. Wskaż tylko wszystkie istotności. Zapis dla rys. 8 w opisie będzie mieć tylko jeden *. Jeśli jednak jest inaczej, prosimy o uwzględnienie tego w Metodzie.

Wyniki dla Rys. 8 - Czy korelacja Pearsona jest korelacją rang? Mam wątpliwości. Należy policzyć współczynnik korelacji Spearmana.

Linia 162. "Łącznie 1 311 618 efektywnych sekwencji..." Zmienianie zapisu Wartość jako matematycznie liczba całkowita, nieoddzielona przecinkami. Inaczej Notacja wskazuje wartości oddzielone przecinkami, co prawdopodobnie nie jest prawdą.

W materiale i metody musisz uzupełnić notację, że w wynikach nazwy poszczególne gatunki drzew będą oznaczone skrótami PM, CC, .....

W tabeli 1 stwierdzić, że obliczane jest również odchylenie standardowe. W materiale i metodach Tego też brakuje.

Reviewer 2 Report

Comments and Suggestions for Authors

Line 24 - TK and TK correct to TP and TK

Line 100 - over 1.1 lakh tons of carbon (?)

Line 120 – icluding correct to including

Line 121 - Hortiboletu, Helotiale, correct to Hortiboletus, Helotiales,

Line 125 – Could not find “ECM fungal species richness” in “Supplemental Table S3”.

Line 127 - Figure 1. Morphological features of different ECM fungi. It will improve the figure 1 if in legend is provided information about which tree species are associated with the ECM species. The host plant is as important as the fungi. Improve the quality of photos to allow to see the bars.

Line 139 - Figure 2. EMF diversity… ECM diversity?

Line 154 – Remove the paragraph

Line 189 - … all samples did not differ significantly.

Line 157 - Russula aff. Kansaiensis (remove the capital letter)

Line 210 - Pseudolabrys. Replace the period with a comma.

Line 226 - Figure 7. … for soil properties…

Line 281 – Remove large space after soils.

Line 279 – Correct Hüblová.

Line 281 – authors of reference 35 should start by Dawud et al.

Line 286 - …neutral (remove the full stop)

Line 329 – Therefore, We (remove capital letter)

Line 402 - .detailed (remove the full stop)

Line 468 – EMF correct to ECM.

Line 572 – Correct the authors names … Joly, F.X.; Hättenschwiler, S.; Vesterdal, L.

The document with “Supplementary Files” must be improved. Correct some text errors like “Supplmental Table 4”, EMF (=ECM). Venn diagrams are out of the order.

Reviewer 3 Report

Comments and Suggestions for Authors

The article is scientifically relevant. It is well written and all its sections are congruent among them. Methodologically, it is robust.

However, authors must pay attention to some points:

i) The taxonomy/systematics of plants, fungi and microorganisms are changing quite fast; thus, it is important to check the valid scientific names in different platforms: ipni.org, GBIF, Tropicos, etc. The complete scientific name includes: Genus species Authority (Family); only genus and species in italics.

ii) The first time a scientific name appears in the text, it ought to be completely written: Genus species Authority/Authorities (Family). If it appears later, only write the first letter of the generic name; i.e. Pinus massoniana Lamb. (Pinaceae); 2nd...n time: P. massoniana. The use of tables facilitates to list all the scientific names, including the Authority/Authorities and the family.

iii) Figures and tables have a Title and an explanatory legend. Readers should completely understand them without the need of reading the whole article or a section of it; in the legend, authors can highlight an important point. As a note, all the abbreviations ought to be explained.

iv) In the References section, authors need to standardize the use of capital and small letters in the title of the articles cited, and the journals´ name.

v) There are minor comments through the text.

Please, see file attached.
